# The Relationship between Quality of Life and Physical Exercise with Depression and Perceived Stress during the Second COVID-19 Lockdown in Greece

**Pavlos Kolias** [1,*] and **Ourania Pliafa** [2]

1 Section of Statistics and Operational Research, Department of Mathematics, Aristotle University of Thessaloniki, 54124 Thessaloniki, Greece
2 Department of Psychology, University of Northampton, Northampton NN1 5PH, UK
* Correspondence: pakolias@math.auth.gr; Tel.: +30-23-1531-7627

**Abstract:** Anxiety, depression, and psychological stress were the most common mental health issues that surfaced during and after the COVID-19 lockdowns. The aim of this paper is to investigate the psychological impact of the second COVID-19 lockdown on the Greek population. A cross-sectional anonymous study was designed, which measured perceived stress, depression symptoms, physical activity, and quality of life. The sample was collected during the period of the second lockdown and consisted of 330 adult individuals (219 females and 111 males) with a mean age of 34.3 years, who were located in Greece. Four scales were applied to measure the constructs, the Perceived Stress Scale (PSS-14), Beck's Depression Inventory (BDI), Quality of Life-BREF (WHOQOL-BREF), and the International Physical Activity Questionnaire (IPAQ). The main findings suggest that depression and perceived stress were more prevalent during the second lockdown compared to previous time periods. The psychological impact was elevated for women and younger individuals. Unemployed people dealt with more stress compared to full-time working individuals. The two quality-of-life domains, physical and psychological health, were negatively correlated with perceived stress and depression. We suggest targeted interventions in order to support the most vulnerable groups and enhance their well-being.

**Keywords:** COVID-19; depression; perceived stress; lockdown; Greece

## 1. Introduction

COVID-19 is a virus that spread vastly and suddenly all over the world, and which, due to its high hospital-admissions and mortality rates at the time, was the cause of abrupt lockdowns, which in turn prompted declines in economic activity and mental health. As the world was unprepared for this health emergency, governments mishandled health departments, economic matters, media information, and eliminated personal freedom for the greater good, universally leading people to an unknown, struggling, lonely, stressed, and/or depressed state [1].

Since the virus's onset, there have been many millions of confirmed COVID-19 cases, deaths due to COVID-19, and recoveries, which on the one hand show promising recovery rates, but on the other hand, the numbers themselves project the magnitude of the pandemic's global impact. Several studies suggest that due to this massive outbreak and its consequences, mental health diseases have either started or already existing ones deteriorated [2–4]. Initial evidence suggested that anxiety and depression symptomatology and self-reports on stress are the most usual psychological repercussions of the COVID-19 pandemic and could also be associated with sleep disturbances [4]. Moreover, various literature reviews support that COVID-19 pandemic was significantly associated with low levels of mental health and psychological distress universally, and in some individuals reached clinical levels [5,6]. Anxiety, depression, posttraumatic stress disorder (PTSD),

psychological stress, and distress are the most common mental health issues that surfaced during and after the lockdowns, mostly due to uncertainty, fear of health and economic instability, loss of a loved one, intense exposure to adversities via media, and many other COVID-19-related topics that affected people [5,6]. Longer quarantines due to higher infection statistics resulted in intense fear of infection, grievance, weariness, insufficient supplies and information, financial hardships, and mostly stigma [2].

Studies concerning several countries assert that in the beginning of the pandemic, citizens from different countries showed high stress levels due to fear of infection, or being at high risk. The isolation itself proved to be a strong stress indicator, and, along with traumatic media exposure, hospitalization, or death of a loved one, and personal symptomatology triggered stress symptoms in extreme levels, which in turn can exacerbate PTSD and depression [7]. A study concerning the Chinese population explained that as China was the first country that was severely affected by the COVID-19, its data on psychological issues such as anxiety, depression, and sleep disturbances were astonishing [8]. China was also the country with the strictest lockdowns and extreme levels of freedom deprivation and propaganda, resulting in the highest levels of generalized anxiety disorder (GAD), depressive symptoms, and sleep quality. The same study showed that the percentages for these three mental health issues were 35.1% for GAD, 20.1% for depression, and 18.2% for sleep disturbances in a large sample. Younger adults showcased the highest prevalence of GAD and depressive symptoms compared to older adults, and the sleep-disturbance data were mostly significant for healthcare workers. Finally, the time spent exposed to COVID-19 news proved to be a significant mediator for GAD [8].

Other countries, such as Austria, presented alarming increased data on anxiety symptoms and depressive symptoms during COVID-19 compared to prior epidemiological data [3]. A considerable percentage (16%) of the sample that was collected, were also over the cut-off point for moderate/severe insomnia. Further, research indicated that the most affected age group was younger adults (<35 years), which inspired this study's focus. Again, significant factors were shown to be low income and job loss, especially with regard to women, and another factor that influenced the direction of this study was the physical inactivity which operated interdependently with stress, depression, and insomnia. A study conducted in Italy on the same matter asserted very similar prevalence of depression, anxiety, and insomnia; they also investigated high perceived stress, adjustment disorder, and posttraumatic stress disorder (PTSD), and the incidence reported was 21.8%, 22.9%, and 37%, respectively, showing that the pandemic's effects on mental health is severe [9].

Concerning the Greek population, studies showed that they were no exception, and that the pandemic affected them negatively at an almost identical rate compared to other countries. Various studies across the country demonstrated findings which show that Greeks were severely affected during and after the first and the second lockdown, in many mental health areas [10–12]. High levels of COVID-19-related fear (35.7%), moderate/severe symptoms of depression (22.8%), and an important percentage of moderate/severe anxiety (77.4%) are only a few of the COVID-19-caused mental health issues [12]. Additionally, women also reported higher scores than men, while younger adults showed less COVID-19-related fear, low depressive symptomatology, and low rates of social responsibility, compared to other countries. COVID-19-related fear was highly associated with a significant other's illness, psychiatric medication intake, and inadequate compliance with guidelines. Further on, age, gender, depression, and anxiety were correlated with COVID-19-related fear. A similar study showed that, although women were more traumatized than men by social distancing and quarantine curfews, men exhibited higher levels of post-traumatic (PT) growth, (life appreciation and spirituality) than women [11]. Stress and post-traumatic stress (PTS) were highly associated with PT growth and enabled its expansion in the context of COVID-19. Finally, the COVID-19 lockdown was associated with high rates of anxiety and depression in senior high-school students [10].

The impact of physical activity (PA) on mental health was also considered in many Greek-population studies, as it provided significant and illuminating results. Although

mandatory home quarantines and isolation measures were correlated with a decrease in physical activity and exercise, these results are obtained from very specific periods of lockdown [13]. As the authors mentioned, in every quarantine phase the overall PA was decreased, however, combined data showed that, towards the quarantine's end, PA scores started to gradually increase again.

The prementioned COVID-19-related mental health issues seemed to manifest other physical abnormalities. Global statistics showed an increase in heart problems, various cancers, substance abuse, and even skin abnormalities. Statistics show that an acne-affected adults have higher self-efficacy in controlling stressful circumstances than others who suffer from other skin disorders. Furthermore, they assert that adults who have skin problems associate the symptoms' worsening with intrapersonal factors, such as fear, stress, and environmental influences [14]. Finally, longitudinal research suggested that perceived stress and PTSD were at higher levels during quarantine, compared to the end of it. Loneliness and use of nonadaptive coping mechanism also had higher levels, whereas resilience, social support, and use of adaptive coping skills were significantly lower. At the time of both lockdowns, PTSD was predicted by loneliness, perceived stress, coping skills, and reduced resilience. Evaluating the pandemic as a crisis also increased PTSD, it was also correlated with female gender, younger age, being single, and childlessness [15].

Considering the above, further research upon the matter is crucial in order to examine any limitations of previous research, ongoing issues, and the possible long-term consequences on public mental health caused by COVID-19 lockdowns. This study, by utilizing a sample of Greek citizens, aims to examine the prevalence of stress and depression symptoms, and their association with demographic characteristics, physical activity and overall quality-of-life.

## 2. Methods

### 2.1. Design and Data Collection

A cross-sectional online study was designed to examine the psychological characteristics and physical activity of Greek population during the period of the second lockdown in Greece, which started on the 7 November 2020. The research has been carried out in accordance with The Code of Ethics of the World Medical Association (Declaration of Helsinki). An anonymous online survey was constructed and made available through Google Forms on the 30 November 2020, and the sample was recruited through social media until the 25 February 2021, approximately three months after the beginning of the second lockdown established by the local authorities. Participants were informed about the aim of the study, the data collection process, and data storage, and they were asked to give consent to provide their anonymous information. The participation was voluntary and fully anonymous, and participants were given the option to withdraw from the survey at any point before submitting their answers.

### 2.2. Participants

The inclusion criteria in the present study were as follows: (a) age over 18 years old, (b) being able to communicate in Greek, and (c) currently located in Greece. The characteristics of the sample ($N = 330$) are presented in Table 1.

**Table 1.** Demographic Characteristics of the Sample.

| Characteristics | N | % |
|---|---|---|
| Gender | | |
| Female | 219 | 66.4 |
| Male | 111 | 33.6 |
| Age | | |
| Mean | 34.31 | |
| Standard deviation | 12.4 | |

**Table 1.** *Cont.*

| Characteristics | *N* | % |
|---|---|---|
| Education level | | |
|     Secondary | 86 | 26.1 |
|     Technical education | 38 | 11.5 |
|     BSc | 133 | 40.3 |
|     MSc/PhD | 73 | 22.1 |
| Working condition | | |
|     Unemployed | 127 | 38.5 |
|     Part-time | 34 | 10.3 |
|     Full-time | 135 | 40.9 |
|     Suspended | 34 | 10.3 |
| Relocation due to lockdown | | |
|     Yes | 45 | 13.6 |
|     No | 285 | 86.4 |
| Currently a university student | | |
|     Yes | 129 | 39.1 |
|     No | 201 | 60.9 |
| Number of children | | |
|     0 | 224 | 67.9 |
|     1 | 33 | 10.0 |
|     2 | 56 | 17.0 |
|     3+ | 17 | 5.1 |

### 2.3. Materials

The following instruments were selected, they are translated into the Greek language, they have been validated, and have also been widely used to measure psychological symptoms in the Greek population.

### 2.4. Perceived Stress (PSS-14)

The translated version of the Perceived Stress Scale (PSS-14) was used to estimate the psychological stress due to COVID-19 and the lockdown [16]. The original version of the scale consists of a one-dimensional instrument with 14 questions on a 5-point Likert scale (0 = Never, 1 = Almost Never, 2 = Sometimes, 3 = Fairly Often, 4 = Very Often), that aims to measure "the degree of general stressful events that take place in participants' life [17]. The PSS-14 has been extensively tested in terms of validity and reliability, showing adequate interval consistency and test–retest reliability [16]. PSS-14's criterion validity was strongly correlated exclusively with the mental-health-status component when measured via 'Medical Outcomes', and finally, the hypothesis testing showed that the PSS-14 was moderately and/or strongly correlated with the emotional variables that were hypothesized, i.e., depression or anxiety [18]. Considering the translated version in Greek, a study suggested that the reliability analysis showed that the interval consistency was estimated to equal $\alpha = 0.82$ and the thresholds of PSS-14 scales were based on previous established cutoffs in the Greek population [16].

### 2.5. Beck Depression Inventory (BDI)

The translated version of the widely used Beck's Depression Inventory (BDI) was implemented to measure depressive symptomatology [19]. The BDI consists of 21 questions, which are answered in a 4-point Likert scale, ranging from "0= Totally Negative Statement" to "3= Totally Positive Statement", with some examples being: "0= I don't feel I am being punished, 1= I feel I may be punished, 2= I expect to be punished, 3= I feel I am being punished". Due to the wide range of the possible answers, the inventory manages to report the depressive symptomatology in the current state of the individual, as it is used for mild and moderate, but also severe cases. The highest score a participant could achieve was 60 (9th question regarding suicidal ideation was omitted for ethical reasons and due to the anonymity of the research) and zero the lowest.

*2.6. Quality of Life-BREF (WHOQOL-BREF)*

The translated version of the WHOQOL-BREF was used to measure the quality of life (QoL), which consists of 24 items and two questions about the general quality of life. The items are divided into four domains: physical health (DOM1), psychological health (DOM2), social relationships (DOM3) and environment (DOM4), during the past two weeks. WHOQOL-BREF has been identified as a cross-culturally reliable and valid instrument for measuring the QoL. WHOQOL-BREF has good to excellent psychometric properties of reliability and performs well in preliminary tests of validity [3]. The internal consistency of the measurement showed a satisfactory Cronbach's α values ranging from 0.78 to 0.90, an adequate test-retest reliability, and discriminant validity [20,21].

*2.7. International Physical Activity Questionnaire (IPAQ)*

For the evaluation of PA, the translated version of the International Physical Activity Questionnaire (IPAQ-GR) was used [22]. The Greek version of the instrument has shown adequate reliability and validity properties in the Greek population [23,24]. The instrument includes four long (31-item) and four short (9-item) questionnaire versions which have been designed to be self-administered or answered by telephone interview. The recall period used by the IPAQ-GR formats is the previous seven days. The purpose of IPAQ-GR is to examine the intensity or volume of PA over the past weekly period and estimate a total PA score expressed in metabolic equivalent task (MET). The instrument attained adequate reliability on the Greek population [23,24].

*2.8. Statistical Analysis*

Frequency tables with counts and percentages were used to describe sample's demographic characteristics. The state of mental health, quality of life, and physical activity for each category of gender was reported as mean and standard deviation. Independent samples *t*-test was used to compare the perceived stress, depression, quality-of-life, and physical activity between the genders. Spearman's correlation coefficient was calculated to provide the relationship between the variables. For the two outcome variables (depression symptoms and perceived stress), two stepwise linear regression models based on the AIC value were estimated using quality-of-life, physical activity, and all the measured demographic characteristics. The aim of our study was to examine the effect of quality-of-life and physical activity as predictors of depression and perceived-stress, adjusting for the demographic characteristics; hence, the stepwise multiple regression model was an appropriate choice in order to provide a subset of candidate variables that adequately fit with the data. K-means clustering was implemented to find an efficient solution of clusters, in order to classify participants based on their perceived stress, depression, quality of life, and physical activity. In order to choose the optimal number of clusters, the elbow method along with the measure of average silhouette width (ASW) was used. The elbow method graphically presents the sum-of-squares error (SSE) in respect with the number of clusters, where the plateau of the graph would indicate the optimal cluster number [25]. The ASW method measures how well subjects fit within each cluster, where higher values indicate more consistent clusters [26]. The significance level was set to a = 0.05 for all statistical tests. The statistical analysis was implemented in R programming language [27].

### 3. Results

Previous findings regarding the prevalence of depression symptoms and perceived stress in the Greek population are presented in Table 2. In our study, the findings suggest that perceived stress during the second lockdown was elevated compared to previous periods, and participants exhibited at least a moderate level of stress in 48.8% of the total sample, while 14.8% of the participants displayed excessive stress. Most of the participants (80.6%) had at least mild symptoms of depression, indicating that depression symptomatology was more prominent during the second lockdown. Table 3 presents the descriptive statistics of depression, perceived stress, quality-of-life, and physical activity across males

and females. Female participants had elevated perceived stress and depression symptoms compared to male participants, with the difference in depression being significant ($t = 2.24$, $p = 0.026$). Females also had higher scores in the social relationships (DOM3) and environment (DOM4) dimensions, while males had higher scores in physical health (DOM1) and psychological health (DOM2) dimensions, although the differences did not attain statistical significance.

**Table 2.** Prevalence of Depression Symptoms and Perceived Stress in Greece.

| Year | Prevalence [1] | Construct | Study | Population |
|------|------------|-----------|-------|-----------|
| 1978 | 17 | Depression (CES-D) | [28] | General |
| 1984 | 27.6 | Depression (CES-D) | [28] | General |
| 2008 | 34.7 | Depression (BDI) | [29] | University students |
| 2011 | 43.9 | Depression (BDI) | [30] | University students |
| 2020 | 80.2 | Depression (BDI) | [31] | University students |
| 2020 | 64.1 | Depression (PHQ-9) | [12] | General |
| 2021 | 63.8 | Depression (PHQ-9) | [10] | High-school students |
| 2022 | 46.6 | Depression (BDI) | [32] | Nursing staff |
| 2020 | 14.64 (7.09) | Perceived stress (PSS-10) | [33] | Nursing staff |
| 2021 | 25.52 (6.07) | Perceived stress (PSS-14) | [14] | Young adults |
| 2021 | 26.46 (7.27) | Perceived stress (PSS-14) | [34] | General |
| 2022 | 28.7 (7.0) | Perceived stress (PSS-10) | [29] | Teaching staff |
| 2022 | 21.32 (6.56) | Perceived stress (PSS-10) | [15] | General |

[1]: Depression symptoms (mild or above) are expressed as percentages (%); perceived stress is expressed with mean values and standard deviations—*M* (*SD*).

**Table 3.** Descriptive Statistics of the Dependent Variables Across Genders.

| | Male | | Female | | |
|------|------|------|--------|------|------|
| | *M* | *SD* | *M* | *SD* | *p* [1] |
| PSS | 28.45 | 6.41 | 29.65 | 5.73 | 0.084 |
| BDI | 23.24 | 13.32 | 26.63 | 12.79 | 0.026 |
| DOM1 | 25.12 | 5.85 | 24.87 | 4.95 | 0.690 |
| DOM2 | 19.37 | 5.55 | 19.18 | 4.42 | 0.740 |
| DOM3 | 9.68 | 3.33 | 10.07 | 3.28 | 0.302 |
| DOM4 | 26.47 | 5.77 | 26.92 | 5.49 | 0.490 |
| MET | 3502.49 | 13453.19 | 2296.05 | 3901.28 | 0.219 |

[1] Independent samples *t*-test comparison with Bonferroni's adjustment.

Table 4 presents the Spearman's correlation coefficient between the outcome variables. Perceived stress had a significant positive correlation with depression symptoms ($r_s = 0.671$, $p < 0.01$), and negative significant correlation with all the quality-of-life dimensions. Depression showed negative significant correlations with all the quality-of-life dimensions, and also exhibited negative correlation with physical activity ($r_s = -0.150$, $p < 0.01$).

**Table 4.** Spearman's Correlation Coefficient Between the Study's Variables.

| | PSS | BDI | DOM1 | DOM2 | DOM3 | DOM4 | MET |
|------|------|------|------|------|------|------|------|
| PSS | - | | | | | | |
| BDI | 0.671 ** | - | | | | | |
| DOM1 | −0.553 ** | −0.698 ** | - | | | | |
| DOM2 | −0.633 ** | −0.720 ** | 0.728 ** | - | | | |
| DOM3 | −0.472 ** | −0.565 ** | 0.611 ** | 0.689 ** | - | | |
| DOM4 | −0.402 ** | −0.512 ** | 0.638 ** | 0.562 ** | 0.520 ** | - | |
| MET | −0.086 | −0.150 ** | 0.141 * | 0.138 * | 0.095 | 0.096 | - |

** $p < 0.01$, * $p < 0.05$.

For the perceived stress, the results of the stepwise regression model based on the AIC are presented in Table 5. The model was overall significant and explained approximately

46.1% of PSS's variability, $F(8, 321) = 36.23$, $p <.001$. The variables that were retained in the model were the gender, age, occupation status, university student status, and physical (DOM1) and psychological health (DOM2). More specifically, full-time working participants showed less perceived stress compared to unemployed participants ($b = -1.95$, $t = 3.345$, $p = 0.001$), while age had a negative effect on perceived stress, with older individuals perceiving less stress compared to those younger ($b = -0.08$, $t = 3.56$, $p < 0.001$). Finally, both physical and psychological health exhibited a negative association with perceived stress.

**Table 5.** Stepwise Linear Regression Model for Perceived Stress (PSS).

| Predictors | Estimate (SE) | *t* | *p* |
|---|---|---|---|
| Intercept | 47.962 (1.337) | 35.869 | <0.001 |
| Gender | | | |
|   Male (Reference) | - | - | - |
|   Female | 0.733 (0.517) | 1.418 | 0.157 |
| Age | −0.083 (0.023) | −3.555 | <0.001 |
| Occupation | | | |
|   Unemployed (Reference) | - | - | - |
|   Part-time | 0.272 (0.868) | 0.314 | 0.754 |
|   Full-time | −1.951 (0.583) | −3.345 | 0.001 |
|   Suspended | −0.936 (0.852) | −1.098 | 0.273 |
| University student | | | |
|   Yes | - | - | - |
|   No | 0.955 (0.594) | 1.607 | 0.109 |
| DOM1 | −0.219 (0.067) | −3.249 | 0.001 |
| DOM2 | −0.552 (0.074) | −7.462 | <0.001 |
| R-squared | 46.1% | | |
| F-statistic (8, 321) | 36.23 | | <0.001 |

The stepwise regression model regarding the depression symptoms (Table 6) explained approximately 61.8% of BDI's variability, $F(5, 324) = 107.4$, $p < 0.001$. The variables that were retained in the model were the gender, the age, the physical (DOM1), psychological (DOM2), and social relationships (DOM3) dimensions. Females had significantly elevated scores on depression compared to males ($b = 2.831$, $t = 2.992$, $p = 0.003$). Similar to perceived stress, the age of the participants had a negative effect on depression ($b = -0.177$, $t = 4.816$, $p < 0.001$). Finally, both physical and psychological health exhibited a negative significant relationship with depression, while social relationship was also negatively associated with depression, although not attaining statistical significance.

**Table 6.** Stepwise Linear Regression Model for Depression Symptoms (BDI).

| Predictors | Estimate (SE) | *t* | *p* |
|---|---|---|---|
| Intercept | 73.699 (2.453) | 30.043 | <0.001 |
| Gender | | | |
|   Male (Reference) | - | - | - |
|   Female | 2.831 (0.946) | 2.992 | 0.003 |
| Age | −0.177 (0.037) | −4.816 | <0.001 |
| DOM1 | −0.847 (0.126) | −6.701 | <0.001 |
| DOM2 | −1.035 (0.152 | −6.830 | <0.001 |
| DOM3 | −0.296 (0.191) | −1.546 | 0.123 |
| R-squared | 61.8% | | |
| F-statistic (5, 324) | 107.4 | | <0.001 |

K-means clustering was implemented to group participants based on their perceived stress, depression, quality of life, and physical activity. The highest value of ASW was

obtained for two clusters, which confirmed the elbow plot, indicating that the optimal number of clusters was k = 2, where the plateau was observed. The continuous characteristics of participants were compared across the two clusters using the Wilcoxon signed-rank test and are presented as medians and lower and upper quartiles (Q1 and Q3), while for the categorical characteristics, the chi-square test was used (Table 7). The first cluster consisted mainly of young female participants (71.3%), who were either unemployed (45.9%) or full time employed (34.4%) and had elevated values in depression symptoms and perceived stress. The second cluster included older individuals (*Mdn* = 37 years), who were working full-time (46.8%), with higher physical activity, elevated scores in all quality-of-life dimensions and decreased symptoms in depression and stress.

**Table 7.** Comparison Of Demographic and Clinical Characteristics Across the Clusters.

| | Cluster 1 (*N* = 157) | Cluster 2 (*N* = 173) | *p* |
|---|---|---|---|
| Gender | | | 0.068 |
| Male | 45 (28.7%) | 66 (38.2%) | |
| Female | 112 (71.3%) | 107 (61.8%) | |
| Age | 27.00 (23.00, 39.00) | 38.00 (26.00, 47.00) | <0.001 |
| Working condition | | | 0.03 |
| Unemployed | 72 (45.9%) | 55 (31.8%) | |
| Part-time | 13 (8.3%) | 21 (12.2%) | |
| Full-time | 54 (34.4%) | 81 (46.8%) | |
| Suspended | 18 (11.4%) | 16 (9.2%) | |
| Relocation due to lockdown | | | 0.073 |
| Yes | 27 (17.2%) | 18 (10.4%) | |
| No | 130 (82.8%) | 155 (89.6%) | |
| PSS | 33.00 (30.00, 36.00) | 26.00 (23.00, 29.00) | <0.001 |
| BDI | 36.00 (31.00, 43.00) | 15.00 (10.00, 21.00) | <0.001 |
| DOM1 | 21.00 (19.00, 25.00) | 29.00 (26.00, 31.00) | <0.001 |
| DOM2 | 16.00 (14.00, 18.00) | 22.00 (20.00, 25.00) | <0.001 |
| DOM3 | 8.00 (6.00, 10.00) | 12.00 (9.00, 14.00) | <0.001 |
| DOM4 | 24.00 (21.00, 28.00) | 29.00 (26.00, 32.00) | <0.001 |
| MET | 916.00 (330.00, 2118.00) | 1470.00 (748.50, 2940.00) | 0.002 |

Continuous variables appear as medians and lower and upper quartiles—Mdn (Q1, Q3); Categorical variables appear as frequencies and percentages—*N* (%).

## 4. Discussion

The aim of the study was to examine the prevalence of the stress and depression symptoms, and their association with demographic characteristics, physical activity, and overall quality-of-life in the Greek population. The main findings indicate that stress and depression were elevated during the second COVID-19 lockdown. Depressive symptoms and stress were negatively associated with the physical and psychological quality-of-life of the individuals. Social relationships, which were restricted during the quarantine, also attained a negative relationship with depression symptoms. Physical activity was also negatively correlated with depression symptoms, yet the impact was not significant while adjusting for the other individual characteristics. Certain demographic characteristics, such as being young, female, and unemployed, were linked to higher scores in depression and perceived stress.

An extensive literature exists that suggest that the quarantine restrictions due to the COVID-19 pandemic had an impact on the psychological health of the individuals [3,7,8,14,15]. The additive effect of the lockdown is also apparent, with previous longitudinal studies showing an increasing trend in anxiety and perceived stress during the second lockdown compared to the first [15]. We have observed greater perceived stress compared to studies in the same country [10,11]. In the second lockdown, reduced physical activity has also been observed compared to previous studies, which evaluated the MET in the first lockdown [13].

In contrast with studies which examined the prevalence of depression and stress before the pandemic, our findings suggest that perceived stress and depression symptoms were elevated during the period of the second lockdown in Greece [16,31,35]. Furthermore, depression symptoms were more apparent to females or unemployed individuals. These findings confirm earlier results in Austria, Italy, Greece, and China, which suggested that female individuals are more affected by depressive symptoms and anxiety than males [3,12,36,37]. This is not a novel finding, as major pandemics and economic crises have a disproportionately negative impact on vulnerable groups, such as minorities, individuals who lack socioeconomic resources, the elderly, and those with chronic conditions [38]. People of low socio-economic status have higher risk of exposure to COVID-19, are not privileged to work remotely, experience financial uncertainty, and often do not have access to health services, with all these factors negatively affecting mental health [38,39].

In the present study, younger individuals had higher scores on both depression and perceived stress, displaying a negative effect of age. On the contrary, a previous study suggested that younger individuals in Greece had lower depression levels and lower fear-related indicators towards COVID-19 [12]. In line with our findings, previous results have shown that sedentariness negatively affects well-being, while physical activity is a positive predictor of well-being [40].

*Limitations*

As this study had a cross-sectional design, it only captured limited information on the psychological impact in one timeframe and comparison with unrestricted periods was not possible. Also, a causal effect between the lockdown and the increase of stress or depression symptoms cannot be directly inferred. We propose a more thorough investigation of the psychological effect of the lockdown, by directly comparing psychological indices via longitudinal designs. Furthermore, the self-report instruments that were used, in addition with the on-line questionnaire deployed, may introduce sample bias.

## 5. Conclusion and Recommendations

We suggest that our findings provide valuable information for targeted interventions and improvement of social welfare. A pandemic is an emergency, and authorities should protect the groups that are more vulnerable. Policymakers should address the vulnerabilities of the most economically disadvantaged within the society and reduce social inequalities and discrimination. Females, and younger unemployed individuals should be the first priority for psychological support. Countries may take preventive measures in order to manage the pandemic; however, all these restrictions should not lead to social isolation. Furthermore, authorities should protect, but not isolate, the people living in institutional settings, and develop targeted measures for addressing the special needs of vulnerable groups, as well as provide health information for those who do not have access to the common communication channels. Finally, targeted interventions that increase mental resilience may support individuals to cope with adverse conditions.

**Author Contributions:** Conceptualization, P.K. and O.P.; Data curation, P.K.; Formal analysis, P.K. and O.P.; Investigation, P.K. and O.P.; Methodology, P.K.; Resources, P.K.; Software, P.K.; Validation, P.K. and O.P.; Writing—original draft, P.K. and O.P.; Writing—review & editing, P.K. and O.P.; visualization, P.K. and O.P.; supervision, P.K.; project administration, P.K. All authors have read and agreed to the published version of the manuscript.

**Funding:** This research received no external funding.

**Institutional Review Board Statement:** The study was conducted in accordance with the Declaration of Helsinki. Ethical review and approval were waived for this study due to its voluntary nature and anonymity.

**Informed Consent Statement:** Informed consent was obtained from all subjects involved in the study. Written informed consent has been obtained from the participants to publish this paper.

**Data Availability Statement:** Data is contained within the article.

**Acknowledgments:** The authors would like to thank Melissa Theocharidou and Despina Dimopoulou for their valuable comments and suggestions.

**Conflicts of Interest:** The authors declare no conflict of interest.

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
