# Peer review of "The Relationship between Quality of Life and Physical Exercise with Depression and Perceived Stress during the Second COVID-19 Lockdown in Greece"

_psych, doi:10.3390/psych4030042_

Round 1
Reviewer 1 Report
Thank you for your contributions with this very relevant topic.
The literature review is thorough and well-cited. The review documents international trends of both mental and physical health challenges resulting from covid-19. The use of a cross-sectional design is appropriate, and the results are presented. The title is appropriate and descriptive.
One recommendation for revision is to align the results tables with the corresponding narrative description. The manuscript currently has the tables and descriptions out of sync, and it is a bit confusing. Also please delete or revise the Patent section.
Author Response
We would like to thank the reviewer for the valuable comments. The tables were aligned with the narrative description, as requested. Also, the Patent section was removed, as it was not necessary for this study.
Reviewer 2 Report
The problem discussed in the article is current and interesting. It is well based on the conclusions of a review of other studies. The aim is clear. The study group and psychological tests are correctly described. It has not been explained why certain variables were included in the regression model and others were omitted. This needs to be completed. Cluster analysis is not related to the aim of this study. It is not clear what this analysis is for.
Author Response
We would like to thank the reviewer for the valuable comments. We have provided the rationale for using a stepwise regression model, that is to find an optimal subset of the predictors, that adequately explain the variability in depression and perceived stress. The cluster analysis aimed to examine if the participants could be divided into homogeneous categories in terms of depressive symptoms, perceived stress, physical activity, and quality of life and also examine whether those categories are different in terms of demographic characteristics and the response variables. With this approach, we expanded the previous regression models and compared the demographic characteristics of the response variables combined.
Reviewer 3 Report
The paper is well written and the results of interest.
The conceptual background could be enlarged to models of vulnerability influencing mental health and well-being.
The practical implications could be illustrated with some more detailed examples.
Author Response
We would like to thank the reviewer for the insightful comments. We have provided a description and two references that denote vulnerability as a key factor that negatively affects physical and mental health. Also, we have added a paragraph with more detailed examples of recommendations.